# New Perspectives in the Pathophysiology and Treatment of Pain in Patients with Dry Eye Disease

**DOI:** 10.3390/jcm11010108

**Published:** 2021-12-25

**Authors:** Giuseppe Giannaccare, Carla Ghelardini, Alessandra Mancini, Vincenzo Scorcia, Lorenzo Di Cesare Mannelli

**Affiliations:** 1Department of Ophthalmology, University Magna Graecia of Catanzaro, 88100 Catanzaro, Italy; alessandra.mancini@studenti.unicz.it (A.M.); vscorcia@unicz.it (V.S.); 2Department of Neuroscience, Psychology, Drug Research and Child Health–NEUROFARBA–Pharmacology and Toxicology Section, University of Florence, 50139 Florence, Italy; carla.ghelardini@unifi.it (C.G.); lorenzo.mannelli@unifi.it (L.D.C.M.)

**Keywords:** dry eye disease, pain, neuropathic pain, opiorphin, glicopro

## Abstract

Ocular discomfort and eye pain are frequently reported by patients with dry eye disease (DED), and their management remains a real therapeutic challenge for the Ophthalmologist. In DED patients, injury at the level of each structure of the ocular surface can determine variable symptoms, ranging from mild ocular discomfort up to an intolerable pain evoked by innocuous stimuli. In refractory cases, the persistence of this harmful signal is able to evoke a mechanism of maladaptive plasticity of the nervous system that leads to increased pain responsiveness. Peripheral and, subsequently, central sensitization cause nociceptor hyperexcitability and persistent pain perception that can culminate in the paradoxical situation of perceiving eye pain even in the absence of ocular surface abnormalities. Effective therapeutic strategies of these cases are challenging, and new options are desirable. Recently, a theoretical novel therapeutic approach concerns enkephalins thanks to the evidence that eye pain sensations are modulated by endogenous opioid peptides (enkephalins, endorphins and dynorphins). In this regard, new topical agents open up a new theoretical scenario in the treatment of ocular discomfort and eye pain in the setting of DED, such as, for example, a multimolecular complex based on proteins and glycosaminoglycans also containing opiorphin that may assist the physiological pain-relieving mechanism of the eye.

## 1. Introduction

Dry eye disease (DED) is a multifactorial condition occurring due to reduced tear production (hyposecretive DED), excessive tear evaporation (evaporative DED) or both (mixed DED). This disorder is characterized by a loss of homeostasis of the tear film in which instability and hyperosmolarity of the tear film itself, inflammation, as well as neurosensory abnormalities play an important role in the development and the maintenance of the disease [1]. DED is characterized by symptoms such as burning, photophobia, blurred vision, and eye discomfort or pain; it is divided into primary or secondary DED, depending on whether it is present in an isolated form or in association with other diseases, mostly autoimmune, such as systemic lupus erythematosus, rheumatoid arthritis, scleroderma, or Sjögren’s syndrome [2]. The pathophysiological mechanism that determines the onset of DED is a change in the quantity and/or composition of tears, which become richer in solutes (tear hyperosmolarity) due to reduced production of the aqueous component secreted by the main lacrimal gland, or its excessive evaporation. The increase in osmolarity, in turn, causes damage to the epithelial cells of the conjunctiva and cornea, as well as to the goblet cells that produce the normal mucous component of tears, and induces an inflammatory reaction at the level of the entire ocular surface [3]. These alterations trigger a vicious circle that aggravates the condition of DED, and determines the chronicization of the process, with damage of the nerve fibers that conduct the stimuli to the main lacrimal gland for the production of the tear film [3]. Recently, the non-invasive imaging of corneal nerves has become possible thanks to the introduction of in vivo confocal microscopy that revealed profound alterations in patients with DED compared to control subjects (Figure 1) [4,5].

In the context of DED, it is of particular importance not only to improve the clinical signs of the disease (e.g., corneal epithelial damage) but also to control symptoms that, in some cases, represent a very invalidating complication for the patient. In fact, a study on health-related quality of life (HR-QoL) of patients with DED has shown that severe forms of the disease affect HR-QoL in a manner comparable to hospital dialysis and severe forms of angina [6]. Furthermore, a more recent study stated that worse mental HR-QoL is present in severe DED patients not having received a clear diagnosis [7].

The management of DED symptoms still remains a great unmet therapeutic need in Ophthalmology and new advances in this field are desirable. The recent advent of a new lubricating ophthalmic solution containing GlicoPro^®^ (Lacricomplex^®^, FB Vision Spa, Ascoli Piceno, Italy) opens up a new theoretical scenario in the treatment of ocular discomfort and eye pain occurring in the setting of DED. GlicoPro^®^ is a multimolecular complex extracted from Helix aspersa snail mucus and based on proteins and sulfured and unsulfured glycosaminoglycans (GAGs), which is carried by a mucin base consisting of hydroxypropyl methylcellulose. GAGs have been shown to be essential for maintaining corneal homeostasis, epithelial cell differentiation, and wound healing. In addition, more recently, a role has been suggested for the extracellular matrix in regulating limbal stem cells, corneal innervation, corneal inflammation, and corneal angiogenesis and lymphangiogenesis. The simple GAGs confer to the GlicoPro^®^ solution a lubricating, moisturizing, antioxidant, and protective action, by reintegrating the mucinic component of the tear film [8]. Sulfur GAGs are thiomers that form covalent bonds (disulfide bridges) with the cysteine residues of mucin. This property makes the GlicoPro^®^ solution highly mucoadhesive and capable of forming the glycocalyx structure in a prolonged manner [9]. Therefore, GlicoPro^®^ has a triple mucomimetic component important for the lubrication of the ocular surface, the stabilization of the tear film, and the prolonged pre-corneal stay. In the protein kit of GlicoPro^®^ there is opiorphin, which assists the physiological pain-relieving mechanism of the eye.

The aim of this review is to summarize the emerging role of opiorphin in the control of ocular discomfort and eye pain.

## 2. Ocular Discomfort and Eye Pain

Symptoms of ocular discomfort or eye pain referred by patients with DED are sustained by various phenomena, such as ocular surface epithelial damage, inflammation, and neurosensory abnormalities. The consequence is the presence of variable symptoms for each single patient, ranging from a mild sensation of ocular discomfort (often reported by the patient as a “foreign body sensation”) to an intolerable pain evoked by innocuous stimuli under normal conditions (allodynia following light, wind, etc.) [10]. The procrastination of this harmful signal is able to evoke a mechanism of maladaptive plasticity of the nervous system that leads to increased pain responsiveness. Peripheral and, subsequently, central sensitization cause nociceptor hyperexcitability and persistent pain perception that can culminate in the paradoxical situation of eye pain, even in the absence of frank ocular surface abnormalities [11,12].

The rupture of the tear film during the blink interval, the hyperosmolarity of the tears, the rubbing between the eyelid and the ocular globe in the presence of decreased tear volume or reduced expression of mucins on the ocular surface, as well as the presence of inflammatory mediators participate in the activation of corneal sensitive fibers capable of triggering nociceptive mechanisms. In fact, the cornea is among the most densely innervated tissues of the body, possessing exclusively C and Ad fibers, comprising mechanonociceptors, polymodal nociceptors (activated by mechanical, chemical, and thermal stimuli), and nociceptors activated by cold stimuli that carry the information to the respective cell bodies located in the trigeminal ganglion projecting to the brainstem nuclei [13,14]. Corneal neurosensory alterations play a fundamental role in both the development and the maintenance of DED. From the initial inflammatory phase, it is possible to evolve toward a neurogenic inflammation that is the basis of real nerve damage with a reduction in the density and branching of nerve fibers up to the formation of neuromas [15]. These phenomena can lead to the onset of a neuropathic type of pain, different in clinical manifestations, amplified in intensity, and persistent over time. The dynamic response of the altered nervous tissue evokes mechanisms of peripheral and central sensitization with increased electrical activity of neurons, excitatory dysregulation of the neurotransmitter milieu, and activation of glial cells that actively participate in synaptic signaling. Neuroplasticity makes this pain central (although originating in the periphery), coupled with a disinhibition of descending pathways, resulting in the loss of the mechanisms that physiologically underlie pain reduction [16]. As the neuropathic component increases, the description of hot, burning, stinging, or granular eye sensation increases as well. Failure of common treatments used for DED is a further indicator of the neuropathic component; patients with little or no relief after the use of conventional treatment based on tear substitutes are often the same ones who report high levels of burning, pain, and wind sensitivity, as well as a decreased threshold for systemic pain [17]. Therapeutic strategies of this challenging condition include: (i) intense lubrication of the ocular surface with tear substitutes in order to, among others, decrease the hyperosmolarity of tears and the consequent stimulation of corneal nociceptors [18]; (ii) contact lens bandaging; (iii) anti-inflammatory agents, such as corticosteroids or immunomodulators, such as cyclosporine or lifitegrast, that can reduce inflammation and increase tear stability; (iv) serum and platelet derivatives, which are rich in growth factors and can promote epithelial wound healing and nerve regeneration [19,20]; and (v) analgesics or anti-neuropathic drugs, such as antiepileptics or antidepressants, for systemic use [17]. This type of persistent pain cannot be controlled by chronically using topical anesthetics because, as reported by Harnish et al., these drugs induce damage to corneal cell metabolism with rarefaction of microvilli and increased cytoplasmic degeneration [21].

Recently, a possible novel therapeutic approach to eye pain concerns enkephalins thanks to the evidence that this symptom is modulated by endogenous opioid peptides (enkephalins, endorphins, and dynorphins) through binding to mu, delta, and kappa opioid receptors widely distributed in the nervous system [22]. The endogenous peptides Met- and Leu-enkephalin, released in the presence of an insult or due to the presence of an inflammatory state by the ocular and corneal nerves, as well as by immune cells (lymphocytes, dendritic cells, and monocytes) recruited at the inflamed site, bind to both mu and delta opioid receptors expressed at the level of the eye [22]. However, enkephalins evoke short-lived local analgesic effects due to their rapid degradation by the concomitant action of two enzymes: neprilysin neutral endopeptidase (NEP) and aminopeptidase N (APN) [23]. On the basis of these aspects, just to overcome the fact that the analgesic action induced by enkephalins is transient, it has been proposed to enhance the effect using compounds that can inhibit the degradation by blocking both enzymes involved (Figure 2).

Reaux-Le Goazigo et al. reported that topical administration of the compound PL265 (a dual inhibitor of the enkephalin-degrading enzymes NEP and APN) appeared to be significantly effective in mouse models of corneal pain [23]. Repeated instillations of this product significantly reduced mechanical and chemical hypersensitivity of the cornea with an action completely antagonized by the opioid antagonist naloxone, highlighting that it was mediated by peripheral corneal opioid receptors. In addition, flow cytometry (quantification of CD11b1 cells) and confocal microscopy analysis revealed that PL265 instillations significantly reduced the active inflammatory process in a model of corneal inflammatory pain by decreasing macrophage infiltrate and expression of the neuronal injury marker ATF3 (Activating Transcription Factor 3) in the ipsilateral trigeminal ganglion. These results, therefore, suggest that the activation of opioid receptors in the eye, through the strategy of raising the concentration of endogenous enkephalins by inhibiting their physiological degradation, could be an effective approach to reduce not only the hyper-nociceptive responses observed in the setting of DED but also the ones following toxic, traumatic, and inflammatory corneal injuries. Molecular and histological studies conducted at the corneal level after repeated instillations of PL265 in a healthy eye have shown that the compound has no detrimental effects on corneal integrity [23]. Furthermore, a recent study suggested that Leu-enkephalin (derived from the same precursor as pro-enkephalin, Met-enkephalin, and prodynorphin) is also able to promote corneal wound repair through the regulation of matrix metalloproteases (MMP-2 and MMP-9) [24]. Finally, regarding the studies about the role of APN/LTA4 (Leukotriene A4) hydrolases (LTA4H) aminopeptidases in extracellular matrix degradation 13, it has been reported that the inhibition of APN/LTA4H activity could prevent the action of extracellular matrix proteases (such as heparanase and MMP-9), thereby slowing down the extracellular matrix degradation process [25]. The prediction of the usefulness of enkephalinase degradation blockers as a new class of topical analgesics, devoid of the side effects of exogenous opioids, is based on the assumption that they merely increase the extracellular physiological concentrations of enkephalins released following stimulus-evoked depolarization. Unlike exogenous opioids that directly stimulate all available opioid receptors, enkephalinase inhibitors act specifically where endogenous enkephalins are released in response to stimulus. On this basis, ocular pain relief by selectively increasing local enkephalin concentrations is likely to be safer than using exogenous opioids in the presence of significant efficacy. A further advantage derives from the absence of an anesthetic effect which, on the contrary, could be harmful for various reasons. On the one hand, corneal anesthesia could conceal the patient’s symptoms in the event of any corneal complications (e.g., erosions, ulcers, etc.), which, although rare, can complicate the course of DED [26]. Moreover, a reduction in corneal sensitivity has already been described in the majority of patients with DED compared to healthy patients of the same age [27]. On the other hand, topical anesthetics, when administered to patients with DED, and the consequent disruption of the corneal epithelium, can lead to an obstacle to the repair of the epithelial defect, alteration of lacrimation, increased corneal permeability with edema and opacification, and, finally, alteration of the elements of the cytoskeleton of the corneal epithelium with alteration of cell motility [28]. These phenomena can lead to chronicity of the corneal epithelial defect and stromal colliquation (melting) up to corneal perforation in the most severe cases [29].

## 3. Opiorphin

Human opiorphin is an endogenous pentapeptide with modulatory capabilities of opioid signaling pathways. It has been identified in human saliva and has potent analgesic properties due to its ability to enhance endogenous opioid signaling by protecting enkephalins from degradation by human NEP and APN [30,31,32]. The opiorphin molecule does not cross the blood–brain barrier due to its intrinsic ability to form bonds with plasma proteins, degree of ionization, and lipid/water partition coefficient. Opiorphin is present in blood, urine, and other body fluids, but because the PROL1 gene, which codes for the opiorphin precursor, is found primarily in human lacrimal and salivary glands, the amount of opiorphin in tears and saliva is higher compared to that one present in other body fluids. Opiorphin is synthesized more in pathological conditions with pain symptomatology [30,31,32]. Salaric et al. reported an increase in salivary opiorphin in patients with burning mouth syndrome [33], while, more recently, Ozdogan et al. described an increase in salivary opiorphin in patients with dental pain caused by pulp inflammation [34]. Opiorphin levels were reported to be increased in the tears of patients with ocular pain caused by a corneal foreign body [35]. This suggests that opiorphin may play a role in the modulation of orofacial and eye pain [8,35]. Opiorphin reduces pain caused by different origins in a manner comparable, if not greater, to morphine in terms of both efficacy and potency [30,36]. Opiorphin exhibits anti-nociceptive effects toward pain induced by thermal stimuli with an opioid receptor-dependent mechanism, and thus the action is lost in the presence of the antagonist naloxone [36,37,38]. In addition, its antihyperalgesic efficacy has been demonstrated in models of alteration of the pain threshold even with neuropathic components, such as following treatment with the neurotoxic substance formalin. Moreover, in this case, the effect of opiorphin is mediated by opioid receptors since the raising of the threshold is antagonized by naloxone [38]. The same authors demonstrated that the effect is selectively μ-receptor dependent as it is blocked by the μ-selective antagonist CTAP, while it is not modified by the K-selective receptor antagonist nor by the δ-selective antagonist naltrindole [38]. Opiorphin has a lower ability to induce constipation, tolerance, and dependence compared to the direct opioid agonist morphine [37,38]. Evidence suggests that after 7 days of treatment, the analgesic effect of opiorphin exceeds the one induced by morphine (at a similar dose), offering a demonstration of the progressive loss of efficacy of the latter and, on the contrary, the maintenance of the effect over time of the former [38]. The greater balance between analgesia and side effects guaranteed by opiorphin is undoubtedly due to its ability to activate opioid pathways by inhibiting the destruction of endogenous enkephalins released in response to a painful stimulus [30,38]. This action has been confirmed by preclinical experiments that reported the achievement of a roof effect (a concentration reached beyond which the effectiveness does not change) [37], highlighting how the action of opiorphin depends on the concentration of enkephalins released following a painful stimulus rather than a direct receptor action independent of physiological regulation and, therefore, easily encroaching on adverse effects [23]. Several studies have shown that other dual NEP and APN enkephalinase inhibitors, such as kelatorphan and thiorphan [39], also induce potent dose-dependent pain suppression in various animal models of pain. However, none of them have been able to produce analgesic effects comparable to those of opiorphin, a greater effectiveness explained by a higher inhibitory capacity of the catabolism of enkephalins [30]. These effects allow a physiological modulation of opioid receptors compared to direct agonists, reducing their side effects (such as respiratory depression, sedation, constipation, physical and psychic dependence, and tolerance). Subchronic treatments with opiorphin do not induce a significant predisposition to abuse, and no dependence phenomena or anti-peristaltic effects have been observed [40]. These aspects represent a crucial aspect as DED has a chronic course characterized by acute episodes of worsening of the symptoms (“poussé”) alternating with periods of partial control, so its treatment is necessarily of long duration.

## 4. Conclusions

DED is a common ocular condition whose symptoms may range from ocular discomfort up to eye pain. To date, the management of chronic ocular pain remains a real therapeutic challenge in Ophthalmology as no specific treatments are effective. GlicoPro^®^ may open up a new theoretical scenario in the treatment of ocular discomfort and eye pain occurring in the setting of DED thanks to its ability to activate opioid pathways by inhibiting the destruction of endogenous enkephalins released in response to a painful stimulus. A recent study has demonstrated the in vitro anti-inflammatory action, the optimal mucoadhesive and regenerative properties, as well as the potential analgesic role of this ocular formulation [41].

However, it should be pointed out that, although the theoretical therapeutic effects in the management of eye pain owing to DED have been described in this paper, in vivo studies on humans are still ongoing and positive results are required in order to support the role of GlicoPro^®^ in the armamentarium of DED therapies.

## Figures and Tables

**Figure 1 jcm-11-00108-f001:**
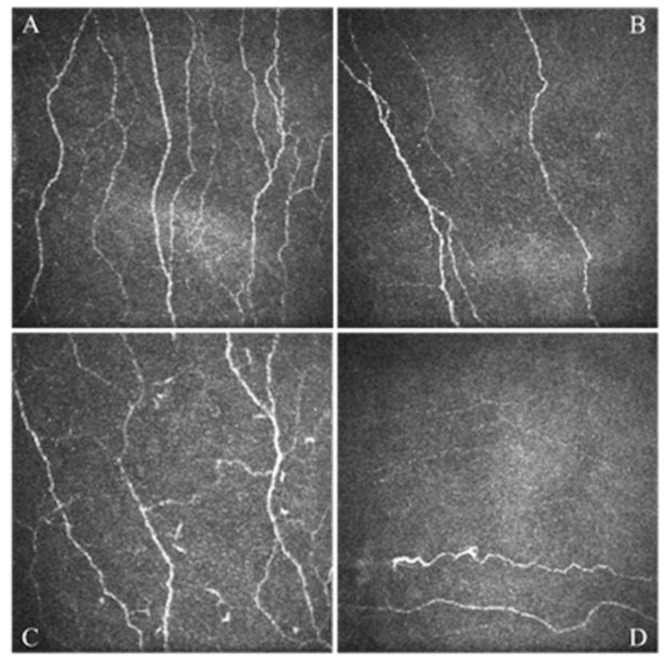
In vivo confocal microscopy scans of the corneal sub-basal nerve plexus obtained in a healthy patient (**A**) and in different patients affected by dry eye disease (**B**–**D**). All images are in the scale of 400 × 400 μm. Part (**A**) shows a normal nerve plexus, part (**B**) shows a nerve plexus with reduced density, part (**C**) shows a nerve plexus with increased tortuosity, part (**D**) shows an altered nerve plexus characterized by the presence of neuroma.

**Figure 2 jcm-11-00108-f002:**
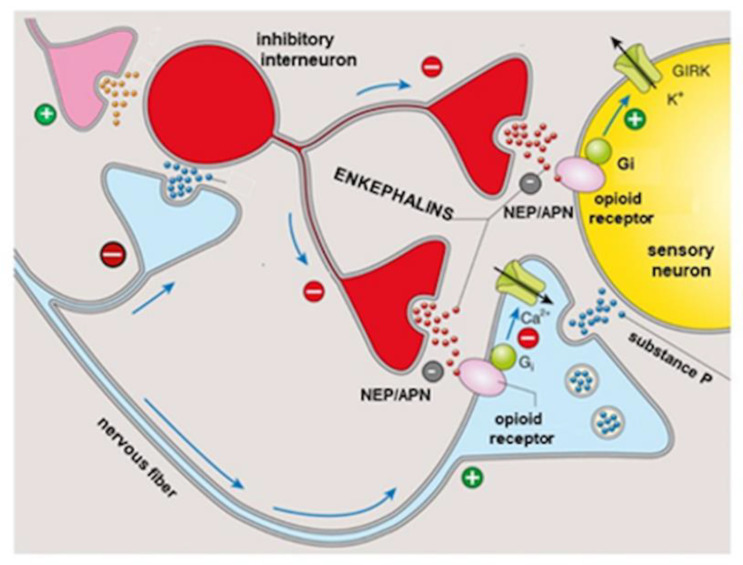
Enkephalin signaling in the pain pathway. Enkephalins are released by inhibitory interneurons activated as a result of the balance between positive (anti-nociceptive) and negative (pain) inputs. Enkephalins are able to activate opioid receptors evoking analgesic signaling; their efficacy is limited by the catabolic enzymes NEP and APN. GIRK = G protein-coupled inwardly rectifying potassium, NEP = neutral endopeptidase, APN = Aminopeptidase N, Gi = Gi protein. Modified from an image available on: www.slideplayer.it. (Accessed on 23 July 2021).

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
