# Peer review of "New Perspectives in the Pathophysiology and Treatment of Pain in Patients with Dry Eye Disease"

_jcm, 2021, doi:10.3390/jcm11010108_

Round 1

Reviewer 1 Report

I accept the revised version.

Author Response

Thanks.

Reviewer 2 Report

The introduction of the paper is a well-written summary of some aspects of DED. The authors refer to the pathophysiology of the neuropathic component of DED. Furthermore, they highlight the meaning of the analgesic characteristics of opiorphin . They report on a possible therapeutic approach to relief the neuratic pain in patients suffering from DED with inhibiors of human NEP and APN

I do not really understand why the authors present "GlicoPro" as a separate (and very short) part no. 4. They do not present any new results or trails. It sounds like a kind of  advertising of this product.

I recommend to include this part into the introduction and not as a separate part. Furthermore, a new paper of Menucci et al. should be cited: Pharmaceutics 202113(12), 2139; https://doi.org/10.3390/pharmaceutics13122139

As I searched for more Information regarding GlicoPro, I found this web site: https://www.helixpharmasrl.com/the-glicopro-inside-the-new-eyedrop-lacricomplex

Is this product licensed? Do you know any approval study on humans?

Author Response

The introduction of the paper is a well-written summary of some aspects of DED. The authors refer to the pathophysiology of the neuropathic component of DED. Furthermore, they highlight the meaning of the analgesic characteristics of opiorphin. They report on a possible therapeutic approach to relief the neuropatic pain in patients suffering from DED with inhibitors of human NEP and APN.

Thanks for your positive comment.

I do not really understand why the authors present "GlicoPro" as a separate (and very short) part no. 4. They do not present any new results or trials. It sounds like a kind of advertising of this product.

I recommend to include this part into the introduction and not as a separate part. Furthermore, a new paper of Mencucci et al. should be cited: Pharmaceutics 202113(12), 2139; https://doi.org/10.3390/pharmaceutics13122139

Thanks for your suggestion to move the part related to GlicoPro at the beginning of the manuscript within the introduction section. Of course, in the revised version we cite also the paper from Prof. Mencucci et al. that has been just published (new ref 41).

As I searched for more Information regarding GlicoPro, I found this web site: https://www.helixpharmasrl.com/the-glicopro-inside-the-new-eyedrop-lacricomplex

Is this product licensed? Do you know any approval study on humans?

GlicoPro is a sterile and non-pyrogenic raw material approved for use in ophthalmic preparations. This raw material is the subject of a patent held by the company Helix Pharma srl located in Ferrara (Italy) and has been exclusively used in Lacricomplex eye drop (FB Vision spa, San Benedetto del Tronto Italy). This product is a class IIb medical device and as such is registered by a notified body through the production of a technical and scientific dossier (data on file). According to the legislation, it is not required to conduct a clinical study before the registration of the product; however, two post-marketing multicenter clinical trials are planned to start soon in Italy.

Reviewer 3 Report

The authors present a thorough and up to date review of pain in dry eye disease, and the potential role of mucins and enkephalins and opiorphin in the modulation of signaling.

  1. In abstract, line 5 “procrastination” is an odd word choice in English. “Persistance” would be a better choice.
  2. The promotion of proprietary agent, in this case GlicoPro, mention in the abstract, is outside the norm for scientific literature. It would be better to simply mention, “new agents, for example, a multimolecular complex . . . “ More info about the agent is presented in the body of the manuscript, which is sufficient. Also, it is unclear in the abstract,  if this new agent is to be used topically or systemically or both.
  3. The authors’ review of the literature on opiorphin is substantial and informative. Unfortunately, their review of role of mucins and GAGs is deficient especially in view of the attention given to the snail mucus product. I think additional attention to the role of mucins and GAGs is warranted.

Author Response

The authors present a thorough and up to date review of pain in dry eye disease, and the potential role of mucins and enkephalins and opiorphin in the modulation of signaling.

  1. In abstract, line 5 “procrastination” is an odd word choice in English. “Persistance” would be a better choice.

Corrected accordingly.

  1. The promotion of proprietary agent, in this case GlicoPro, mention in the abstract, is outside the norm for scientific literature. It would be better to simply mention, “new agents, for example, a multimolecular complex . . . “ More info about the agent is presented in the body of the manuscript, which is sufficient. Also, it is unclear in the abstract,  if this new agent is to be used topically or systemically or both.

Corrected accordingly.

  1. The authors’ review of the literature on opiorphin is substantial and informative. Unfortunately, their review of role of mucins and GAGs is deficient especially in view of the attention given to the snail mucus product. I think additional attention to the role of mucins and GAGs is warranted.

We agree with you that the majority of the information provided in the manuscript are focused on opiorphin rather than on GAGs and mucins. The reason is that the aim of the review is to describe the potential of the new approach mediated by opiorphin for controlling eye pain rather than to describe the characteristics of all the other molecules contained in the product. However, we added some sentences about the role of mucins and GAGs contained in GlicoPro® in the introduction section as follows: “GlicoPro® is a multimolecular complex extracted from Helix aspersa snail mucus and based on proteins and sulfured and unsulfured glycosaminoglycans (GAGs), which is carried by a mucin base consisting of hydroxypropyl methylcellulose. GAGs have been shown to be essential for maintaining corneal homeostasis, epithelial cell differentiation and wound healing, and, more recently, a role has been suggested for the extracellular matrix in regulating limbal stem cells, corneal innervation, corneal inflammation, corneal angiogenesis and lymphangiogenesis. The simple GAGs confer to the GlicoPro® solution a lubricating, moisturizing, antioxidant, and protective action, by reintegrating the mucinic component of the tear film [8]. Sulfur GAGs are thiomers that form covalent bonds (disulfide bridges) with the cysteine residues of mucin. This property makes the GlicoPro® solution highly mucoadhesive and capable of forming the glycocalyx structure in a prolonged manner [9]. Therefore, GlicoPro® has a triple mucomimetic component important for the lubrication of the ocular surface, the stabilization of the tear film, and the prolonged pre-corneal stay.”

This manuscript is a resubmission of an earlier submission. The following is a list of the peer review reports and author responses from that submission.

Round 1

Reviewer 1 Report

The review is very interesting, well and clearly written. It addresses an important issue in dry eye disease physiopathology and treatment: neurosensorial abnormalities and their management.  

Moreover, opiorphin seems a very promising innovative therapeutic approach for dry eye, possibly improving patients’ quality of life.   

Nevertheless, the paragraph 4, which concerns the new ophthalmic formulation, needs to be revised in order to be useful for readers: as it is written now, it seems mere advertising. More specifically:  

- page 6 line 242-244 “The multimolecular principle of action of GlicoPro® results in the relief of symptoms of ocular discomfort and eye pain” and “Figure 3. Features and benefits of using GlicoPro® in dry eye disease”: are there studies which support these statements? If not, please specify that these are possible therapeutic effects of GlicoPro, and also that in vitro and clinical studies are necessary to confirm them.    

 -Which is the suggested administration regimen in order to obtain relief of ocular discomfort and eye pain?  

 -Is Lacricomplex available in all countries? 

-Is GlicoPro fully chemically synthetized or is it extracted from natural sources? 

 Moreover, some passages of the manuscript lack references: i.e., pages 1-2 lines 35-49, page 3 lines 115-128, page 4 lines 148-150, page 8 lines 184-188.  

Author Response

We thank the Editor and the Reviewers for the revision of our work submitted to be considered for publication on Journal of Clinical Medicine. We hope that the helpful comments of the Reviewers have contributed in making our paper clearer and stronger.

Reviewer’ comments are showed in normal text, our replies in bold.

The review is very interesting, well and clearly written. It addresses an important issue in dry eye disease physiopathology and treatment: neurosensorial abnormalities and their management.  

Thanks for your positive comments. As stated in the updated TFOS definition of dry eye disease, neurosensorial abnormalities represent a key feature of the disease and their treatment is still challenging for Ophthalmologists.

Moreover, opiorphin seems a very promising innovative therapeutic approach for dry eye, possibly improving patients’ quality of life.   

Thanks for your positive opinion. We agree with you that this molecule could represent a new fascinating approach for the management of patients with dry eye disease.

Nevertheless, the paragraph 4, which concerns the new ophthalmic formulation, needs to be revised in order to be useful for readers: as it is written now, it seems mere advertising. More specifically:  

- page 6 line 242-244 “The multimolecular principle of action of GlicoPro® results in the relief of symptoms of ocular discomfort and eye pain” and “Figure 3. Features and benefits of using GlicoPro® in dry eye disease”: are there studies which support these statements? If not, please specify that these are possible therapeutic effects of GlicoPro, and also that in vitro and clinical studies are necessary to confirm them.    

Thanks for the comment that allows us to make clearer for the reader the section regarding Lacricomplex. We agree with you that in vivo and in vitro studies on Lacricomplex are required to confirm its potential effects. Therefore, we added as last sentence of the conclusion section the following statement: “Although the possible therapeutic effects of opiorphin in the management of eye pain owing to DED has been widely described in this paper, in vitro and in vivo studies on humans are required in order to confirm these findings.”

Which is the suggested administration regimen in order to obtain relief of ocular discomfort and eye pain?  

The information for the user contained in the package leaflet suggests the posology of 1-2 drops in the conjunctival sac 3-4 times a day. We added this information in the related section of the text, as follows: “The features and potential principles of action of GlicoPro® - instilled 3-4 times a day - in the relief of symptoms of ocular discomfort and eye pain is represented in Figure 3.

Is Lacricomplex available in all countries? 

No, to date it is commercially available only in Italy. The registration process is currently ongoing in other countries.

Is GlicoPro fully chemically synthetized or is it extracted from natural sources?

GlicoPro is extracted from helix aspersa snail mucus and then purified and sterilized. We added this information in the related section of the text, as follows: “Lacricomplex® (FB Vision, Ascoli  Piceno, Italy) is a new lubricating, protective, moisturizing, soothing, and antioxidant ophthalmic solution that contains GlicoPro®, a multimolecular complex extracted from Helix aspersa snail mucus and based on proteins and sulfured and unsulfured glycosaminoglycans (GAGs), which is carried by a mucin base consisting of hydroxypropyl methylcellulose.”

Moreover, some passages of the manuscript lack references: i.e., pages 1-2 lines 35-49, page 3 lines 115-128, page 4 lines 148-150, page 8 lines 184-188.  

We agree with you and added the lacking references accordingly.

Reviewer 2 Report

The manuscript is an introduction of a new eye drop formulation for the treatment of dry eye disease but unfortunately without any supporting real life patient experiences or clinical trial results. The article details the basic science (pharmacological, animal) facts and abdominal and oral human results about the main compound (opiorphin) of a new eye drop formulation and pharmacological theories about the possible actions on the ocular surface. However, no clinical evidences, no clinical trial results were presented which might support the expected actions and expected treatment results in dry eye disease in human. Without clinical data, the expectations are supported only theoretically and thus it is only a theoretical opening of a new way of a possibly promising new treatment scenario.   

The main aim(s) of the present review article is(are) not described at the end of the Introduction.

(No references to the owner and the permission for use of the images in the legend of Figures if these are not made by the authors.)

Author Response

We thank the Editor and the Reviewers for the revision of our work submitted to be considered for publication on Journal of Clinical Medicine. We hope that the helpful comments of the Reviewers have contributed in making our paper clearer and stronger.

Reviewer’ comments are showed in normal text, our replies in bold.

The manuscript is an introduction of a new eye drop formulation for the treatment of dry eye disease but unfortunately without any supporting real life patient experiences or clinical trial results.

Thanks for your comment that allows us to be clearer in the text. We are aware that this paper reviews the potential roles of opiorphin in the treatment of eye pain occurring in dry eye disease, and that in vitro and in vivo studies on humans are required to confirm these findings. Therefore, we added as last sentence of the conclusion section the following statement: “Although the possible therapeutic effects of opiorphin in the management of eye pain owing to DED has been widely described in this paper, in vitro and in vivo studies on humans are required in order to confirm these findings.”

The article details the basic science (pharmacological, animal) facts and abdominal and oral human results about the main compound (opiorphin) of a new eye drop formulation and pharmacological theories about the possible actions on the ocular surface. However, no clinical evidences, no clinical trial results were presented which might support the expected actions and expected treatment results in dry eye disease in human. Without clinical data, the expectations are supported only theoretically and thus it is only a theoretical opening of a new way of a possibly promising new treatment scenario.   

As above, we agree with you that in vitro and in vivo studies on humans are required to confirm the theoretical mechanisms of action that are summarized in the paper. As suggested, we added the term “theoretical” to the scenario opened by this molecule.

The main aim(s) of the present review article is(are) not described at the end of the Introduction.

Yes, we agree with you and we added the following sentences at the end of the introduction section: "Since the management of DED symptoms still remains a great unmet therapeutic need in Ophthalmology, new advances in this field are desirable. The aim of this review is to summarize the role of opiorphin in the control of (eye) pain, and to further describe the features and the potential benefits of a new topical product cointaining opiorphin for the control of DED symptoms.”

No references to the owner and the permission for use of the images in the legend of Figures if these are not made by the authors.

Figures 1 and 3 are original; figure 2 was modified from an image available on “www.slideplayer.it". This information was now added in the revised manuscript.

Round 2

Reviewer 2 Report

The authors responded to the first review adequately. I accept the review of the scientific knowledge on the topic in the present form.

However, the presentation of a new eye drop product, Lacricomplex® (at the end of the abstract and in chapter 4. of the article) is still misleading without supportive human clinical data. At least there is a need to mention this limitation directly in connection with the product, as well. Or alternatively, deletion of product description from the abstract and from the text might be a possible solution.

Author Response

Reviewer’ comments are showed in normal text, our replies in bold.

The authors responded to the first review adequately. I accept the review of the scientific knowledge on the topic in the present form.

Thanks for your positive comment.

However, the presentation of a new eye drop product, Lacricomplex® (at the end of the abstract and in chapter 4. of the article) is still misleading without supportive human clinical data. At least there is a need to mention this limitation directly in connection with the product, as well. Or alternatively, deletion of product description from the abstract and from the text might be a possible solution.

We agree with you and added a clear statement in the final part of the section about Lacricomplex® where we explained the need for clinical trial data and real life ones. The new statement is: It should be highlighted that Lacricomplex® has just been introduced in the market and clinical trial data as well as real life ones are not yet available. These results are needed to support the efficacy of Lacricomplex® and its role in the armamentarium of DED therapies.”